# Affinity Clustering: Hierarchical Clustering at Scale

**MohammadHossein Bateni**
Google Research
bateni@google.com

**Soheil Behnezhad**[*]
University of Maryland
soheil@cs.umd.edu

**Mahsa Derakhshan**[*]
University of Maryland
mahsaa@cs.umd.edu

**MohammadTaghi Hajiaghayi**[*]
University of Maryland
hajiagha@cs.umd.edu

**Raimondas Kiveris**
Google Research
rkiveris@google.com

**Silvio Lattanzi**
Google Research
silviol@google.com

**Vahab Mirrokni**
Google Research
mirrokni@google.com

## Abstract

Graph clustering is a fundamental task in many data-mining and machine-learning pipelines. In particular, identifying a good hierarchical structure is at the same time a fundamental and challenging problem for several applications. The amount of data to analyze is increasing at an astonishing rate each day. Hence there is a need for new solutions to efficiently compute effective hierarchical clusterings on such huge data.

The main focus of this paper is on minimum spanning tree (MST) based clusterings. In particular, we propose *affinity*, a novel hierarchical clustering based on Borůvka's MST algorithm. We prove certain theoretical guarantees for affinity (as well as some other classic algorithms) and show that in practice it is superior to several other state-of-the-art clustering algorithms.

Furthermore, we present two MapReduce implementations for affinity. The first one works for the case where the input graph is dense and takes constant rounds. It is based on a *Massively Parallel* MST algorithm for dense graphs that improves upon the state-of-the-art algorithm of Lattanzi *et al.* [34]. Our second algorithm has no assumption on the density of the input graph and finds the affinity clustering in $O(\log n)$ rounds using Distributed Hash Tables (DHTs). We show experimentally that our algorithms are scalable for huge data sets, e.g., for graphs with trillions of edges.

## 1   Introduction

Clustering is a classic unsupervised learning problem with many applications in information retrieval, data mining, and machine learning. In hierarchical clustering the goal is to detect a nested hierarchy of clusters that unveils the full clustering structure of the input data set. In this work we study the hierarchical clustering problem on real-world graphs. This problem has received a lot of attention in recent years [13, 16, 41] and new elegant formulations and algorithms have been introduced. Nevertheless many of the newly proposed techniques are sequential, hence difficult to apply on large data sets.

---

[*]Supported in part by NSF CAREER award CCF-1053605, NSF BIGDATA grant IIS-1546108, NSF AF:Medium grant CCF-1161365, DARPA GRAPHS/AFOSR grant FA9550-12-1-0423, and another DARPA SIMPLEX grant.

With the constant increase in the size of data sets to analyze, it is crucial to design efficient large-scale solutions that can be easily implemented in distributed computing platforms (such as Spark [45] and Hadoop [43] as well as MapReduce and its extension Flume [17]), and cloud services (such as Amazon Cloud or Google Cloud). For this reason in the past decade several papers proposed new distributed algorithms for classic computer science and machine learning problems [3, 4, 7, 14, 15, 19]. Despite these efforts not much is known about distributed algorithms for hierarchical clustering. There are only two works analyzing these problems [27, 28], and neither gives any theoretical guarantees on the quality of their algorithms or on the round complexity of their solutions.

In this work we propose new parallel algorithms in the MapReduce model to compute hierarchical clustering and we analyze them from both theoretical and experimental perspectives. The main idea behind our algorithms is to adapt clustering techniques based on classic minimum spanning tree algorithms such as Borůvka's algorithm [11] and Kruskal's algorithm [33] to run efficiently in parallel. Furthermore we also provide a new theoretical framework to compare different clustering algorithms based on the concept of a "certificate" and show new interesting properties of our algorithms.

We can summarize our contribution in four main points.

First, we focus on the distributed implementations of two important clustering techniques based on classic minimum spanning tree algorithms. In particular we consider linkage-based clusterings inspired by Kruskal's algorithm and a novel clustering called *affinity clustering* based on Borůvka's algorithm. We provide new theoretical frameworks to compare different clustering algorithms based on the concept of a "certificate" as a proof of having a good clustering and show new interesting properties of both affinity and single-linkage clustering algorithms.

Then, using a connection between linkage-based clustering, affinity clustering and the minimum spanning tree problem, we present new efficient distributed algorithms for the hierarchical clustering problem in a MapReduce model. In our analysis we consider the most restrictive model for distributed computing, called *Massively Parallel Communication*, among previously studied MapReduce-like models [10, 23, 30]. Along the way, we obtain a constant round MapReduce algorithm for minimum spanning tree (MST) of dense graphs (in Section 5). Our algorithm for graphs with $\Theta(n^{1+c})$ edges and for any given $\epsilon$ with $0 < \epsilon < c < 1$, finds the MST in $\lceil \log(c/\epsilon) \rceil + 1$ rounds using $\tilde{O}(n^{1+\epsilon})$ space per machine and $O(n^{c-\epsilon})$ machines (i.e., optimal total space). This improves the round complexity of the state-of-the-art MST algorithm of Lattanzi *et al.* [34] for dense graphs which requires up to $\lceil c/\epsilon \rceil$ rounds using the same number of machines and space. Prior to our work, no hierarchical clustering algorithm was known in this model.

Then we turn our attention to real world applications and we introduce efficient implementations of affinity clustering as well as classic single-linkage clustering that leverage Distributed Hash Tables (DHTs) [12, 31] to speed up computation for huge data sets.

Last but not least, we present an experimental study where we analyze the scalability and effectiveness of our newly introduced algorithms and we observe that, in most cases, affinity clustering outperforms all state-of-the-art algorithms from both quality and scalability standpoints.[2]

## 2   Related Work

Clustering and, in particular, hierarchical clustering techniques have been studied by hundreds of researchers [16, 20, 22, 32]. In social networks, detecting the hierarchical clustering structure is a basic primitive for studying the interaction between nodes [36, 39]. Other relevant applications of hierarchical clustering can be found in bioinformatics, image analysis and text classification.

Our paper is closely related to two main lines of research. The first one focuses on studying theoretical properties of clustering approaches based on minimum spanning trees (MSTs). Linkage-based clusterings (often based on Kruskal's algorithm) have been extensively studied as basic techniques for clustering datasets. The most common linkage-based clustering algorithms are single-linkage, average-linkage and complete-linkage algorithms. In [44], Zadeh and Ben-David gave a characterization of the single-linkage algorithm. Their result has been then generalized to linkage-based algorithms in [1]. Furthermore single-linkage algorithms are known to provably recover a ground truth clustering if the similarity function has some stability properties [6]. In this paper we

introduce a new technique to compare clustering algorithms based on "certificates." Furthermore we introduce and analyze a new algorithm—affinity—based on Borůvka's well-known algorithm. We show that affinity is not only scalable for huge data sets but also its performance is superior to several state-of-the-art clustering algorithms. To the best of our knowledge though Borůvka's algorithm is a well-known and classic algorithm, not many clustering algorithms have been considered based on Borůvka's.

The second line of work is closely related to distributed algorithms for clustering problems. Several models of MapReduce computation have been introduced in the past few years [10, 23, 30]. The first paper that studied clustering problems in these models is by Ene *et al.* [18], where the authors prove that any $\alpha$ approximation algorithm for the $k$-center or $k$-median problems can produce $4\alpha + 2$ and $10\alpha + 3$ approximation factors, respectively, for the $k$-center or $k$-median problems in the MapReduce model. Subsequently several papers [5, 7, 8] studied similar problems in the MapReduce model. A lot of efforts also went into studying efficient algorithms on graphs [3, 4, 7, 15, 14, 19]. However the problem of hierarchical clustering did not receive a lot of attention. To the best of our knowledge there are only two papers [27, 28] on this topic, and neither analyzes the problem formally or proves any guarantee in any MapReduce model.

# 3  Minimum Spanning Tree-Based Clusterings

We begin by going over two famous algorithms for minimum spanning tree and define the corresponding algorithms for clustering.

**Borůvka's algorithm and affinity clustering:** *Borůvka's algorithm* [11], first published in 1926, is an algorithm for finding a minimum spanning tree (MST)[3]. The algorithm was rediscovered a few times, in particular by Sollin [42] in 1965 in the parallel computing literature. Initially each vertex forms a group (cluster) by itself. The algorithm begins by picking the cheapest edge going out of each cluster, in each *round* (in parallel) joins these clusters to form larger clusters and continues joining in a similar manner until a tree spanning all vertices is formed. Since the size of the smallest cluster at least doubles each time, the number of rounds is at most $O(\log n)$. In *affinity clustering*, we stop Borůvka's algorithm after $r > 0$ rounds when for the first time we have at most $k$ clusters for a desired number $k > 0$. In case the number of clusters is strictly less than $k$, we delete the edges that we added in the last round in a non-increasing order (i.e., we delete the edge with the highest weight first) to obtain exactly $k$ clusters. To the best of our knowledge, although Borůvka's algorithm is a well-known and classic algorithm, clustering algorithms based on it have not been considered much. A natural hierarchy of nodes can be obtained by continuing Borůvka's algorithm: each cluster here will be a subset of future clusters. We call this *hierarchical affinity clustering*.

We present distributed implementations of Borůvka/affinity in Section 5 and show its scalability even for huge graphs. We also show affinity clustering, in most cases, works much better than several well-known clustering algorithms in Section 6.

**Kruskal's algorithm and single-linkage clustering:** Kruskal's algorithm [33] first introduced in 1956 is another famous algorithm for finding MST. The algorithm is highly sequential and iteratively picks an edge of the least possible weight that connects any two trees (clusters) in the forest.[4] Though the number of iterations in Kruskal's algorithm is $n - 1$ (the number of edges of any tree on $n$ nodes), the algorithm can be implemented in $O(m \log n)$ time with simple data structures ($m$ is the number of edges) and in $O(ma(n))$ time using a more sophisticated disjoint-set data structure, where $a(.)$ is the extremely slowly growing inverse of the single-valued Ackermann function.

In *single-linkage clustering*, we stop Kruskal's algorithm when we have at least $k$ clusters (trees) for a desired number $k > 0$. Again if we desire to obtain a corresponding *hierarchical single-linkage clustering*, by adding further edges which will be added in Kruskal's algorithm later, we can obtain a natural hierarchical clustering (each cluster here will be a subset of future clusters).

As mentioned above, Kruskal's Algorithm and single-linkage clustering are highly sequential, however as we show in Section 5 thinking backward once we have an efficient implementation of Borůvka's

(or any MST algorithm) in Map-Reduce and using Distributed Hash Tables (DHTs), we can achieve an efficient parallel implementation of single-linkage clustering as well. We show scalability of this implementation even for huge graphs in Section 5 and its performance in experiments in Section 6.

## 4  Guaranteed Properties of Clustering Algorithms

An important property of affinity clustering is that it produces clusters that are roughly of the same size. This is intuitively correct since at each round of the algorithm, each cluster is merged to at least one other cluster and as a result, the size of even the smallest cluster is at least doubled. In fact linkage based algorithms (and specially single linkage) are often criticized for producing uneven clusters; therefore it is tempting to give a theoretical guarantee for the size ratio of the clusters that affinity produces. Unfortunately, as it is illustrated in Figure 1, we cannot give any worst case bounds since even in one round we may end up having a cluster of size $\Omega(n)$ and another cluster of size $O(1)$. As the first property, we show that at least in the first round, this does not happen when the observations are randomly distributed. Our empirical results on real world data sets in Section 6.1, further confirm this property for all rounds, and on real data sets.

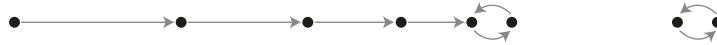

Figure 1: An example of how affinity may produce a large component in one round.

We start by defining the nearest neighbor graph.

**Definition 1** (Nearest Neighbor Graph). *Let $S$ be a set of points in a metric space. The nearest neighbor graph of $S$, denoted by $\mathcal{G}_S$, has $|S|$ vertices, each corresponding to an element in $S$ and if $a \in S$ is the nearest element to $b \in S$ in $S$, graph $\mathcal{G}_S$ contains an edge between the corresponding vertices of $a$ and $b$.*

At each round of affinity clustering, all the vertices that are in the same connected component of the nearest neighbor graph will be merged together[5]. Thus, it suffices to bound the connected components' size.

For a random model of points, consider a *Poisson point process $X$* in $\mathbb{R}^d$ ($d \geq 1$) with density 1. It has two main properties. First, the number of points in any finite region of volume $V$ is Poisson distributed with mean $V$. Second, the number of points in any two disjoint regions are independent of each other.

**Theorem 1** (Häggström *et al.* [38]). *For any $d \geq 2$, consider the (Euclidean distance) nearest neighbor graph $\mathcal{G}$ of a realization of a Poisson point process in $\mathbb{R}^d$ with density 1. All connected components of $\mathcal{G}$ are finite almost surely.*

Theorem 1 implies that the size of the maximum connected component of the points within any finite region in $\mathbb{R}^d$ is bounded by almost a constant number. This is a very surprising result compared to the worst case scenario of having a connected component that contains all the points.

Note that although the aforementioned bound holds for the first round of affinity, after the connected components are contracted, we cannot necessarily assume that the new points are Poisson distributed and the same argument cannot be used for the rest of the rounds.

Next we present further properties of affinity clustering. Let us begin by introducing the concept of "cost" for a clustering solution to be able to compare clustering algorithms.

**Definition 2.** *The cost of a cluster is the sum of edge lengths (weights) of a minimum Steiner tree connecting all vertices inside the cluster. The cost of a clustering is the sum of the costs of its clusters. Finally a non-singleton clustering of a graph is a partition of its vertices into clusters of size at least two.*

Even one round of affinity clustering often produces good solutions for several applications. Now we are ready to present the following extra property of the result of the first round of affinity clustering.

**Theorem 2.** *The cost of any non-singleton clustering is at least half of that of the clustering obtained after the first round of affinity clustering.*

Before presenting the proof of Theorem 2, we need to demonstrate the concept of *disc painting* introduced previously in [29, 2, 21, 9, 25]. In this setting, we consider a topological structure of a graph metric in which each edge is a curve connecting its endpoints whose length is equal to its weight. We assume each vertex has its own color. A *disc painting* is simply a set of disjoint disks centered at terminals (with the same colors of the center vertices). A disk of *radius* $r$ centered at vertex $v$ paints all edges (or portions) of them which are at distance $r$ from vertex $v$ with the color of $v$. Thus we paint (portions of) edges by different disks each corresponding to a vertex and each edge can be painted by at most two disks. With this definition of disk painting, we now demonstrate the proof of Theorem 2.

Next we turn our focus to obtain structural properties for single-linkage clustering. We denote by $F_k$ the set of edges added after $k$ iterations of Kruskal, i.e., when we have $n - k$ clusters in single-linkage clustering. Note that $F_k$ is a forest, i.e., a set of edges with no cycle. First we start with an important observation whose proof comes directly from the description of the single-linkage algorithm.

**Proposition 3.** *Suppose we run single-linkage clustering until we have $n - k$ clusters. Let $d_{\text{outside}}$ be the minimum distance between any two clusters and $d_{\text{inside}}$ be the maximum distance of any edge added to forest $F_k$. Then $d_{\text{outside}} \geq d_{\text{inside}}$.*

We note that Proposition 3 demonstrates the following important property of single-linkage clustering: *Each vertex of a cluster at any time has a neighbor inside to which is closer than any other vertex outside of its clusters.*

Next we define another criterion for desirability of a clustering algorithm. This generalizes Proposition 3.

**Definition 3.** *An $\alpha$-certificate for a clustering algorithm, where $\alpha \geq 1$, is an assignment of* shares *to each vertex of the graph with the following two properties: (1) The cost of each cluster is at most $\alpha$ times the sum of shares of vertices inside the cluster; (2) For any set $S$ of vertices containing at most one from each cluster in our solution, the imaginary cluster $S$ costs at least the sum of shares of vertices in $S$.*

Note that intuitively the first property guarantees that vertices inside each cluster can pay the cost of their corresponding cluster and that there is no free-rider. The second property intuitively implies we cannot find any better clustering by combining vertices from different clusters in our solution.

Next we show that there always exists a 2-certificate for single-linkage clustering guaranteeing its worst-case performance.

**Theorem 4.** *Single-linkage always produces a clustering solution that has a 2-certificate.*

## 5    Distributed Algorithms

### 5.1    Constant Round Algorithm For Dense Graphs

Unsurprisingly, finding the affinity clustering of a given graph $G$ is closely related to the problem of finding its Minimum Spanning Tree (MST). In fact, we show the data that is encoded in the MST of $G$ is sufficient for finding its affinity clustering (Theorem 9). This property is also known to be true for single linkage [24]. For MapReduce algorithms this is particularly useful because the MST requires a substantially smaller space than the original graph and can be stored in one machine. Therefore, once we have the MST, we can obtain affinity or single linkage in one round.

The main contribution of this section is an algorithm for finding the MST (and therefore the affinity clustering) of dense graphs in constant rounds of MapReduce which improves upon prior known dense graph MST algorithms of Karloff *et al.* [30] and Lattanzi *et al.* [34].

**Theoretical Model.** Let $N$ denote the input size. There are a total number of $M$ machines and each of them has a space of size $S$. Both $S$ and $M$ must be substantially sublinear in $N$. In each round, the machines can run an arbitrary polynomial time algorithm on their local data. No communication is allowed during the rounds but any two machines can communicate with each other between the rounds as long as the total communication size of each machine does not exceed its memory size.

---

**Algorithm 1** MST of Dense Graphs

---

**Input**: A weighted graph $G$
**Output**: The minimum spanning tree of $G$

 1: **function** MST($G = (V, E), \epsilon$)
 2:     $c \leftarrow \log_n (m/n)$                    ▷ Since $G$ is assumed to be dense we know $c > 0$.
 3:     **while** $|E| > \mathrm{O}(n^{1+\epsilon})$ **do**
 4:         REDUCEEDGES($G, c$)
 5:         $c \leftarrow (c - \epsilon)/2$
 6:     Move all the edges to one machine and find MST of $G$ in there.
 7: **function** REDUCEEDGES($G = (V, E), c$)
 8:     $k \leftarrow n^{(c-\epsilon)/2}$
 9:     Independently and u.a.r. partition $V$ into $k$ subsets $\{V_1, \ldots, V_k\}$.
10:     Independently and u.a.r. partition $V$ into $k$ subsets $\{U_1, \ldots, U_k\}$.
11:     Let $G_{i,j}$ be a subgraph of $G$ with vertex set $V_i \cup U_j$ containing any edge $(v, u) \in E(G)$
     where $v \in V_i$ and $u \in U_j$.
12:     **for** any $i, j \in \{1, \ldots, k\}$ **do**
13:         Send all the edges of $G_{i,j}$ to the same machine and find its MST in there.
14:         Remove an edge $e$ from $E(G)$, if $e \in G_{i,j}$ and it is not in MST of $G_{i,j}$.

---

This model is called Massively Parallel Communication ($\mathcal{MPC}$) in the literature and is "arguably the most popular one" [26] among MapReduce like models.

**Theorem 5.** *Let $G = (V, E)$ be a graph with $n$ vertices and $n^{1+c}$ edges for any constant $c > 0$ and let $w : E \mapsto \mathbb{R}^+$ be its edge weights. For any given $\epsilon$ such that $0 < \epsilon < c$, there exists a randomized algorithm for finding the MST of $G$ that runs in at most $\lceil \log (c/\epsilon) \rceil + 1$ rounds of $\mathcal{MPC}$ where every machine uses a space of size $\tilde{\mathrm{O}}(n^{1+\epsilon})$ with high probability and the total number of required machines is $\mathrm{O}(n^{c-\epsilon})$.*

Our algorithm, therefore, uses only enough total space ($\tilde{\mathrm{O}}(n^{1+c})$) on all machines to store the input.

The following observation is mainly used by Algorithm 1 to iteratively remove the edges that are not part of the final MST.

**Lemma 6.** *Let $G' = (V', E')$ be a (not necessarily connected) subgraph of the input graph $G$. If an edge $e \in E'$ is not in the MST of $G'$, then it is not in the MST of $G$ either.*

To be more specific, we iteratively divide $G$ into its subgraphs, such that each edge of $G$ is at least in one subgraph. Then, we handle each subgraph in one machine and throw away the edges that are not in their MST. We repeat this until there are only $\mathrm{O}(n^{1+\epsilon})$ edges left in $G$. Then we can handle all these edges in one machine and find the MST of $G$. Algorithm 1 formalizes this process.

**Lemma 7.** *Algorithm 1 correctly finds the MST of the input graph in $\lceil \log (c/\epsilon) \rceil + 1$ rounds.*

By Lemma 6 we know any edge that is removed from is not part of the MST therefore it suffices to prove the while loop in Algorithm 1 takes $\lceil \log (c/\epsilon) \rceil + 1$ iterations.

**Lemma 8.** *In Algorithm 1, every machine uses a space of size $\tilde{\mathrm{O}}(n^{1+\epsilon})$ with high probability.*

The combination of Lemma 7 and Lemma 8 implies that Algorithm 1 is indeed in $\mathcal{MPC}$ and Theorem 5 holds. See supplementary material for omitted proofs.

The next step is to prove all the information that is required for affinity clustering is indeed contained in the MST.

**Theorem 9.** *Let $G = (V, E)$ denote an arbitrary graph, and let $G' = (V, E')$ denote the minimum spanning tree of $G$. Running affinity clustering algorithm on $G$ gives the same clustering of $V$ as running this algorithm on $G'$.*

By combining the MST algorithm given for Theorem 5 and the sufficiency of MST for computing affinity clustering (Theorem 9) and single linkage ([24]) we get the following corollary.

**Corollary 10.** *Let $G = (V, E)$ be a graph with $n$ vertices and $n^{1+c}$ edges for any constant $c > 0$ and let $w : E \mapsto \mathbb{R}^+$ be its edge weights. For any given $\epsilon$ such that $0 < \epsilon < c$, there exists a*

*randomized algorithm for affinity clustering and single linkage that runs in $\lceil \log(c/\epsilon) \rceil + 1$ rounds of $\mathcal{MPC}$ where every machine uses a space of size $\tilde{O}(n^{1+\epsilon})$ with high probability and the total number of required machines is $O(n^{c-\epsilon})$.*

## 5.2 Logarithmic Round Algorithm For Sparse Graphs

Consider a graph $G(V,E)$ on $n = |V|$ vertices, with edge weights $w : E \mapsto \mathbb{R}$. We assume that the edge weights denote distances. (The discussion applies *mutatis mutandis* to the case where edge weights signify similarity.)

The algorithm works for a fixed number of synchronous rounds, or until no further progress is made, say, by reaching a single cluster of all vertices. Each round consists of two steps: First, every vertex picks its *best* edge (i.e., that of the minimum weight) at each round; and then the graph is contracted along the selected edges. (See Algorithm 2 in the appendix.)

For a connected graph, the algorithm continues until a single cluster of all vertices is obtained. The supernodes at different rounds can be thought of as a hierarchical clustering of the vertices.

While the first step of each round has a trivial implementation in MapReduce, the latter might take $\Omega(\log n)$ MapReduce rounds to implement, as it is an instance of the connected components problem. Using a DHT was shown to significantly improve the running time here, by implementing the operation in one round of MapReduce [31]. Basically we have a read-only random-access table mapping each vertex to its best neighbor. Repeated lookups in the table allows each vertex to follow the chain of best neighbors until a loop (of length two) is encountered. This assigns a unique name for each connected component; then all the vertices in the same component are reduced into a supernode.

**Theorem 11.** *The affinity clustering algorithm runs in $O(\log n)$ rounds of MapReduce when we have access to a distributed hash table (DHT). Without the DHT, the algorithm takes $O(\log^2 n)$ rounds.*

# 6 Experiments

## 6.1 Quality Analysis

In this section, we compare well known hierarchical and flat clustering algorithms, such as $k$-means, single linkage, complete linkage and average linkage with different variants of affinity clustering, such as single affinity, complete affinity and average affinity. We run our experiments on several data sets from the UCI database [37] and use Euclidean distance[6].

To evaluate the outputs of these algorithms we use Rand index which is defined as follows.

**Definition 4** (Rand index [40]). *Given a set $V = \{v_1, \ldots, v_n\}$ of $n$ points and two clusterings $X = \{X_1, \ldots, X_r\}$ and $Y = \{Y_1, \ldots, Y_s\}$ of $V$. Define the following.*

- *$a$: the number of pairs in $V$ that are in the same cluster in $X$ and in the same cluster in $Y$.*

- *$b$: the number of pairs in $V$ that are in different clusters in $X$ and in different clusters in $Y$.*

*the Rand index $r(X,Y)$ is defined to be $(a+b)/\binom{n}{2}$. By having the ground truth clustering $T$ of a data set, we define the Rand index score of a clustering $X$, to be $r(X,T)$.*

The Rand index based scores are in range $[0,1]$ and a higher number implies a better clustering. For a hierarchical clustering, the level of its corresponding tree with the highest score is used for evaluations.

Figure 2 (a) compares the Rand index score of different clustering algorithms for different data sets. We observe that single affinity generally performs really well and is among the top two algorithms for most of the datasets (all except Glass). Average affinity also seems to perform well and in some cases (e.g., for Soybean data set) it produces a very high quality clustering compared to others. To summarize, linkage based algorithms do not seem to be as good as affinity based algorithms but in some cases $k$-means could be close.

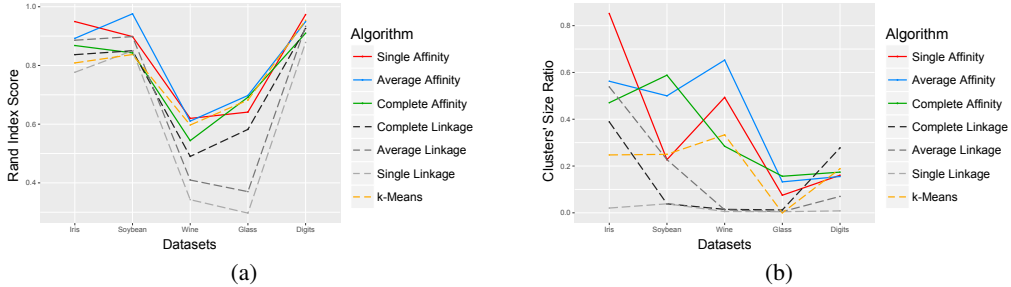

(a)                                                                 (b)

Figure 2: Comparison of clustering algorithms based on their Rand index score (a) and clusters size ratio (b).

Table 1: Statistics about datasets used. (Numbers for ImageGraph are approximate.) The fifth column shows the relative running time of affinity clustering, and the last column is the speedup obtained by a ten-fold increase in parallelism.

| Dataset | # nodes | # edges | max degree | running time | speedup |
|---|---|---|---|---|---|
| LiveJournal | 4,846,609 | 7,861,383,690 | 444,522 | 1.0 | 4.3 |
| Orkut | 3,072,441 | 42,687,055,644 | 893,056 | 2.4 | 9.2 |
| Friendster | 65,608,366 | 1,092,793,541,014 | 2,151,462 | 54 | 5.9 |
| ImageGraph | $2 \times 10^{10}$ | $10^{12}$ | 14000 | 142 | 4.1 |

Another property of the algorithms that we measure is the clusters' *size ratio*. Let $X = \{X_1, \ldots, X_r\}$ be a clustering. We define the size ratio of $X$ to be $\min_{i,j\in[r]} |X_i|/|X_j|$. As it is visualized in Figure 2 (b), affinity based algorithms have a much higher size ratio (i.e., the clusters are more balanced) compared to linkage based algorithms. This confirms the property that we proved for Poisson distributions in Section 4 for real world data sets. Hence we believe affinity clustering is superior to (or at least as good as) the other methods when the dataset under consideration is not extremely unbalanced.

## 6.2 Scalability

Here we demonstrate the scalability of our implementation of affinity clustering. A collection of public and private graphs of varying sizes are studied. These graphs have between 4 million and 20 billion vertices and from 4 billion to one trillion edges. The first three graphs in Table 1 are based on public graphs [35]. As most public graphs are unweighted, we use the number of common neighbors between a pair of nodes as the weight of the edge between them. (This is computed for all pairs, whether they form a pair in the original graph or not, and then new edges of weight zero are removed.) The last graph is based on (a subset of) an internal corpus of public images found on the web and their similarities.

We note that we use the "maximum" spanning tree variant of affinity clustering; here edge weights denote similarity rather than distance.

While we cannot reveal the exact running times and number of machines used in the experiments, we report these quantities in "normalized form." We only run one round of affinity clustering (consisting of a "Find Best Neighbors" and a "Contract Graph" step). Two settings are used in the experiments. We once use $W$ MapReduce workers and $D$ machines for the DHT, and compare this to the case with $10W$ MapReduce workers and $D$ machines for the DHT. This ten-fold increase in the number of MapReduce workers leads to four- to ten-fold decrease in the total running time for different datasets. Each running time is itself the average over three runs to reduce the effect of external network events.

Table 1 also shows how the running time changes with the size of the graph. With a modest number of MapReduce workers, affinity clustering runs in less than an hour for all the graphs.

## Footnotes

[2]Implementations are available at `https://github.com/MahsaDerakhshan/AffinityClustering`.

[3]More precisely the algorithm works when there is a unique MST, in particular, when all edge weights are distinct; however this can be easily achieved by either perturbing the edge weights by an $\epsilon > 0$ amount or have a tie-breaking ordering for edges with the same weights

[4]Unlike Borůvka's method, this greedy algorithm has no limitations on the distinctness of edge weights.

[5]Depending on the variant of affinity that we use, the distance function will be updated.

[6]We consider Iris, Wine, Soybean, Digits and Glass data sets.

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
