[Supplementary Material]

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

# A  Appendix

## A.1  Section 4

*Proof of Theorem 2.* For a vertex $v$ of the graph, let $v_{\text{nearest}}$ be its nearest neighbor in the first iteration of affinity clustering. Now for each vertex $v$ of the graph, we paint a disk of radius $\text{distance}(v, v_{\text{nearest}})/2$ with its color $v$. By definition, all disks will be disjoint (i.e., paint different parts of the graph). As a result, the union of Steiner trees corresponding to any non-singleton clustering must intersect each disk at least to the extend of the disk's radius. Therefore, the total length of the union of Steiner trees is at least $\sum_{v \in V(G)} \text{distance}(v, v_{\text{nearest}})/2$ which is half the cost of the affinity clustering just after the first round. $\qquad\square$

*Proof of Theorem 4.* First we start with the notion of *moat painting* which is a generalization of disk painting described above. A *moat* is a generalization of a disk which surrounds a set of vertices and its color is that of one of the vertices in this set. The *depth of a moat* generalizes the concept of the radius of a disk and is the minimum length that any path should traverse from outside the moat to inside the moat. Bateni, Hajiaghayi, and Marx [9] introduce the concept of *prize-collecting clustering* which itself builds on primal-dual algorithms of Goemans and Williamson [21] and Agrawal, Klein and Ravi [2]. It is also worth mentioning that the idea of disc painting and its intuitive analysis predates the modern primal-dual analysis and it is originally introduced by Jünger and Pulleyblank [29]. Via prize-collecting clustering Bateni *et al.* build a moat painting (with disjoint moats) for clusters generated by single-linkage clustering since single-linkage clustering is a special case of their prize-collecting clustering (with infinite penalties for all vertices). The share of each vertex in our certificate is the sum of depths of the moats surrounding the vertex with the same color as the vertex itself.

Now the factor two in Property 1 of the certificate indeed comes directly from factor two in the approximation guarantee of Bateni et al. [9].

Proof of Property 2 of our certificate is Similar to that of Theorem 2. Disjointness of moats implies that any Steiner tree of a set $S$ of vertices must intersect each moat at least to the extent of its depth, and these moats paint disjoint portions of edges in any Steiner tree of set $S$. Thus the cost of (an imaginary cluster) $S$ is at least the sum of the depths of moats with colors of any vertex in $S$ which in turn is equal to the sum of the shares of the vertices in $S$. This proves Property 2 of our certificate. $\qquad\square$

## A.2  Section 5.1

*Proof of Lemma 6.* It is a known fact that any graph with distinct edge weights has a unique MST. Let $M$ denote the MST of $G$ and let $M'$ denote the MST of $G'$. For the sake of contradiction, assume that there exists an edge $e = (u, v) \in M$ that is not in $M'$. Since $e \notin M'$, there is a path $p$ from $v$ to $u$ in $M'$ where $w(e') < w(e)$ for any $e' \in p$. Removing $e$ from $M$ partitions it into two connected components, one of which contains vertex $u$ and the other one contains vertex $v$. We already know that path $p$ connects $u$ to $v$, so it has at least an edge $e'$ that connects these two components. Since $w(e') < w(e)$, we can replace $e$ with $e'$ in $M$ to obtain a better MST for $G$, which is a contradiction. $\qquad\square$

*Proof of Lemma 7.* Let $c_r$ denote the value of variable $c$ at the $r$-th iteration (round) of the while loop in Algorithm 1. At round $r$, the algorithm independently and uniformly at random partitions the vertices of $G$ into two partition sets $U = \{U_1, \ldots, U_{k_r}\}$, and $V = \{V_1, \ldots, V_{k_r}\}$, where $k_r = n^{(c_r - \epsilon)/2}$. Then, for any $i, j \in \{1, .., k_r\}$, it finds the MST of $G_{i,j}$, which we denote by $T_{i,j}$, and removes any edge in $G_{i,j}$ that is not in $T_{i,j}$. By Lemma 6, none of the removed edges are part of the solution. Therefore if the algorithm terminates, the output is indeed correct.

Now it suffices to prove the algorithm terminates in $\lceil \log{(c/\epsilon)} \rceil + 1$ rounds.

We first prove the total number of edges that will be kept by the end of the $r$-th round is at most $k_r n$. The key observation is that the MST of a subgraph with $n'$ vertices has at most $n' - 1$ edges. This allows us to charge each edge of $T_{i,j}$ to one of the vertices of $G_{i,j}$ such that no two edges of $T_{i,j}$ are charged to the same vertex. Fix any vertex $v$ and assume it is in set $V_i$. Vertex $v$ is charged by at most $k_r$ edges since $v$ is only in $G_{i,1}, G_{i,2}, \ldots, G_{i,k_r}$. This means all the vertices combined are charged by at most $k_r n$ edges and hence $k_r n$ is an upper bound for the number of edges that are left by the end of the $r$-th round. Recall that $k_r = n^{(c_r - \epsilon)/2}$, hence $k_r n = n^{1 + (c_r - \epsilon)/2}$. On the other hand, note that $c_r < c/2^r$ since $c_0 = c$ and $c_r = (c_{r-1} - \epsilon)/2$. This implies that in round $\lceil \log{(c/\epsilon)} \rceil$,

$$c_{\lceil \log{(c/\epsilon)} \rceil} < \frac{c}{2^{\lceil \log{(c/\epsilon)} \rceil}} <= \epsilon.$$

Therefore after $\lceil \log{(c/\epsilon)} \rceil$ rounds at most $O(n^{1+\epsilon})$ edges are left in the graph. This is exactly the termination condition of the while loop. In the final round of the algorithm, we send all the remaining edges to one machine and find the MST of $G$, therefore it takes at most $\lceil \log{(c/\epsilon)} \rceil + 1$ rounds for the algorithm to terminate and correctly report the MST. $\qquad\square$

We first state a lemma that will be useful in proving the forthcoming lemmas.

**Lemma 12.** *Let $\beta$ be a constant number in $(1,2)$, and let $S = \{x_1, \ldots, x_n\}$ be a set of numbers where $\sum_{i=1}^{n} x_i = n^{\beta}$, and $\max_{i=1}^{n} x_i \leq n$. For any two constant numbers $c$ and $\alpha$ where $0 < \alpha \leq b$, if we distribute members of $S$ into $n^{\alpha}$ groups uniformly and independently at random, with probability at least $1 - \frac{\log n}{n^c}$, total sum of the numbers in any group is at most $\tilde{O}(n^{\beta-\alpha})$.*

*Proof.* Let $G_i$ denote the $i$-th group, and let $D_j \subset S$ denote the set of numbers such that for any $x \in D_j$, $2^{j-1} \leq x \leq 2^j$. For any $i$ and $j$ with $j \in \{1, \ldots, \log n\}$ and $i \in \{1, \ldots, n^{\alpha}\}$ we prove that with probability at least $1 - \frac{1}{n^c}$,

$$\sum_{x \in D_j \cap G_i} x \leq \tilde{O}(n^{\beta-\alpha}).$$

It obviously holds for any $j$ where $|D_j| \leq n^{\beta-\alpha-1}$ since $n \cdot n^{\beta-\alpha-1} = n^{\beta-\alpha}$. In addition for any $j$ that $|D_j| > n^{\beta-\alpha-1}$, by a simple application of Chernoff bound, weith probability at least $1 - \frac{1}{n^c}$ for any $G_i$, $|D_j \cap G_i| \leq \frac{c|D_j| \log n}{n^{\alpha}}$. Since $|D_j| \leq O(\frac{n^{\beta}}{2^j})$, and $\sum_{x \in D_j \cap G_i} x \leq 2^j \cdot |D_j \cap G_i|$, we have

$$\sum_{x \in D_j \cap G_i} \deg(v) \leq 2^j \cdot O(\frac{n^{\beta} \log n}{2^j n^{\alpha}}) = \tilde{O}(n^{\beta-\alpha}).$$

Therefore by Union bound with probability at least $1 - \frac{\log n}{n^c}$, for any $G_i$,

$$\sum_{x \in G_i} x = \sum_{j=1}^{\log n} \sum_{x \in D_j \cap G_i} x \leq \sum_{j=1}^{\log n} \tilde{O}(n^{\beta-\alpha}) = \tilde{O}(n^{\beta-\alpha}),$$

concluding the proof. $\square$

*Proof of Lemma 8.* Let $G$ denote the graph in an arbitrary round of the algorithm which has $n^{1+c}$ edges. Note that in any round of the algorithm we have two sets of graph partitions, denoted by $U = \{U_1, \ldots, U_k\}$ and $V = \{V_1, \ldots, V_k\}$, where $k = n^{(c-\epsilon)/2}$. Moreover, $G_{i,j}$ is the subgraph with vertex set $V_i \cup U_i$ that contains all the edges of $G$ with one end-point in $V_i$ and one end-point in $U_j$. To prove this algorithm does not violate the space limits, we need to prove for any $i$ and $j$ that $G_{i,j}$ has at most $\tilde{O}(n^{1+\epsilon})$ edges. We first give a bound for the total degree of vertices in one partition. Then we prove that with high probability the number of edges in any machine is $\tilde{O}(n^{1+\epsilon})$. Let $\deg(v)$ denote the degree of vertex $v$ in graph $G$, and let $\deg_{i,j}(v)$ denote the degree of vertex $v$ in $G_{i,j}$.

Note that $|V| = n^{(c-\epsilon)/2}$, and the total degree of vertices in $G$ is $O(n^{1+c})$. By Lemma 12, for any partition $V_i \in V$, with probability at least $1 - \log n/n^3$,

$$\sum_{v \in V_i} \deg_i(v) = \tilde{O}(n^{1-c}/n^{(c-\epsilon)/2}) = \tilde{O}(n^{1-(c+\epsilon)/2}).$$

Let $\deg(V_i) := \sum_{v \in V_i} \deg(v)$ denote the sum of degrees of all the vertices in $V_i$. Note that for any edge $e \in G$ whose one end point is in $V_i$, there exists exactly one $j \in [k]$ such that $e$ is in $G_{i,j}$. Therefore, $\sum_{j=1}^{k} |E(G_{i,j})| = \deg(V_i)$.

Since $|U| = n^{(c-\epsilon)/2}$, by Lemma 12, with probability at least $1 - \log n/n^3$, for any $j$,

$$|E(G_{i,j})| = \tilde{O}(\frac{\deg(V_i)}{n^{(c-\epsilon)/2}}).$$

If $\deg(V_i) = \tilde{O}(n^{1-(c+\epsilon)/2})$, then with probability at least $1 - \log n/n^3$, for any $j$, $|E(G_{i,j})| = \tilde{O}(n^{1+\epsilon})$ which is the space of each machine. By an application of Union bound, probability of holding $|E(G_{i,j})| = \tilde{O}(n^{1+\epsilon})$ for any $i, j$ is

$$1 - (k+1) \cdot (\log n/n^3) \geq 1 - \log n/n^2.$$

By Lemma 7, the number of rounds of this algorithm is $\lceil \log(c/\epsilon) \rceil + 1$, therefore with probability at least $1 - (\lceil \log(c/\epsilon) \rceil + 1) \cdot \log n/n^2$ (high probability) throughout this algorithm each machine needs a space of at most $\tilde{O}(n^{1+\epsilon})$. $\square$

*Proof of Theorem 9.* For the sake of contradiction assume that running affinity clustering on $G$ gives a different clustering of $V$ than running it on $G'$. For any round $i$, and graph $H$ let $C(i, H)$ denote the set of clusters that we have in round $i$ of running affinity clustering algorithm on graph $H$. It is easy to see that $C(0, G') =$

---

**Algorithm 2** Affinity Clustering

---

**Input**: A graph $G = (V, E)$
**Output**: One clustering (denoted by mapping $\lambda_i$) per level

1: $i \leftarrow 0$
2: **repeat**
3:      Find Best Neighbors for $G$ to get $\lambda$            ▷ See Algorithm 3 in the appendix.
4:      Contract Graph $G$ based on $\lambda$ to get $G'$     ▷ See Algorithm 4 in the appendix.
5:      $G \leftarrow G'$
6:      $\lambda_i \leftarrow \lambda$
7:      $i \leftarrow i + 1$
8: **until** $\lambda$ is the identity mapping

---

**Algorithm 3** Find Best Neighbors

---

**Input**: A graph $G = (V, E)$
**Output**: A mapping $\lambda : V \mapsto V$
Run the following MapReduce where $N(v)$ is the set of neighbors of $v \in V$.
**Map** $\langle u; N(u) \rangle$

1: **if** $N(u) = \emptyset$ **then**
2:      Emit $\langle u; u \rangle$
3: **else**
4:      $v \leftarrow \arg\min_{v \in N(u)} w(u, v)$
5:      Emit $\langle u; v \rangle$

**Reduce** $\langle u; v \rangle$

1: Emit $\langle u; v \rangle$

---

$C(0, G) = V$. Let $j$ denote the first round that $C(j, G') \neq C(j, G)$. This means that there exists at least one cluster $c_1$ in both $C(j - 1, G')$ and $C(j - 1, G)$ that its cheapest edge going out of the cluster in $G$, denoted by $e_1$, is different from its cheapest edge going out of the cluster in $G'$, denoted by $e_2$. We know $E' \subset E$, therefore $w(e_2) > w(e_1)$. Let $e_1 = (u, v)$. Since $G'$ is an MST of $G$ there is a path $p$ in $G'$ from $u$ to $v$ such that any edge in this path is cheaper than $e_1$ (otherwise we can replace $e_1$ with an edges of $p$ in $G'$ and decrease the cost of the MST which is a contradiction). Existing a path between $u$ and $v$ with all the edges cheaper than $e_1$, implies that there is an edge going out of cluster $c_1$ in $E'$ that is cheaper than $e_1$ which is a contradiction since $E' \subset E$ and $e_1$ is the cheapest edge going out of the cluster $c_1$ in $G$. $\qquad\square$

## A.3    Section 5.2

Algorithm 2 shows the MapReduce implementation using DHTs. To implement the algorithm purely in MapReduce without a DHT, we only need to replace the Map phase of Algorithm 4 (see the appendix) with the connected-components algorithm developed in prior work [31].

*Proof of Theorem 11.* Affinity clustering (Algorithm 2) makes $O(\log n)$ calls to "Find Best Neighbors" (Algorithm 3) and "Contract Graph" (Algorithm 4), because each round reduces the number of vertices by a factor of at least two. "Find Best Neighbors" has a simple logic and takes only one round of MapReduce. "Contract Graph," on the other hand, is a special case of the connected components problem, that may be implemented in $O(\log n)$ rounds of MapReduce [31] without a DHT, or in one round using a DHT as shown in Algorithm 4. As a result, affinity clustering takes $O(\log n)$ and $O(\log^2 n)$ rounds of MapReduce with and without a DHT, respectively. $\qquad\square$

**Algorithm 4** Contract Graph

---

**Input**: A graph $G = (V, E)$ and mapping $\lambda : V \mapsto V$
**Output**: A contracted graph $G' = (V', E')$
Run the following MapReduce with random access via a DHT to $\lambda$.
**Map** $\langle u; N(u) \rangle$

 1: $c \leftarrow v \leftarrow u$
 2: $S \leftarrow \emptyset$
 3: **while** $v \notin S$ **do**
 4:     $S \leftarrow S \cup \{v\}$
 5:     $c \leftarrow \min(c, v)$
 6:     $v \leftarrow \lambda(v)$
 7: Emit $\langle c; N(u) \rangle$

**Reduce** $\langle u; F = \{A_1, A_2, \ldots, A_l\} \rangle$

 1: $N \leftarrow \bigcup_{A \in F} A$
 2: Emit $\langle u; N \rangle$

---