[Reviews · NeurIPS 2017]

Reviewer 1



The paper focuses on the development of the field of distributed hierarchical clustering. The authors propose a novel class of algorithms tagged 'affinity clustering' that operate on the basis of Boruvka's seminal work on minimal spanning trees and contrast those to linkage clustering algorithms (which are based on Kruskal's work). The authors systematically introduce the theoretical underpinnings of affinity clustering, before proposing 'certificates' as a metric to characterise clustering algorithm solutions more generally by assessing the clustered edge weights (cost). Following the theoretical analysis and operationalisation of MapReduce variants of affinity clustering for distributed operation, the quality is assessed empirically using standard datasets with variants of linkage- and affinity-based algorithms, as well as k-means. In addition to the Rand index (as metric for clustering accuracy) the quality of algorithms is assessed based on the ratio of the detected clusters (with balanced cluster sizes considered favourable). Affinity-based algorithms emerge favourably for nearly all datasets, with in parts significant improvements (with the flat clustering algorithm k-means as closest contestant). Finally, the scalability of affinity clustering is assessed using private and public corpi, with near-linear scalability for the best performing case. Overall, the paper proposes a wide range of concepts that extend the field of hierarchical clustering (affinity-based clustering, certificates as QA metric). As far as I could retrace, the contributions are systematically developed and analysed, which warrants visibility. One (minor) comment includes the experimental evaluation. In cases where the affinity-based algorithms did not perform as well (an example is the cluster size ratio for the Digits dataset), it would have been great to elaborate (or even hypothesise) why this is the case. This would potentially give the reader a better understanding of the applicability of the algorithm and potential limitations. Minor typos/omissions: - Page 8, Table 1: "Numbered for ImageGraph are approximate." - Page 8, Subsection Scalability: Missing reference for public graphs.

Reviewer 2



The paper provides a demonstration of MapReduce based hierarchical clusterings. The basic premise is that the edge-removal based algorithms (namely, the heaviest edge in a cycle cannot be in an MST) can be parallelized in MapReduce, specially if we can use certificates (of primal-dual optimality) to compare across different graphs. The experiment scales are corresponding improvements are noteworthy and that is the strong point of the paper. The low point of the paper is the obfuscation/confoundedness in theory. One wishes that the paper said the summary that was laid out above - the two sentences likely convey more information that the authors wrote. First of all, the authors should refer to disc paintings and certificates by their original notions such as region growing and primal-dual algorithms. It may be the case that [9,24] managed to get papers published without mentioning that these ideas were invented elsewhere, but the authors should realize that getting a paper accepted does not avoid the issue of being laughed at by posterity. And in case the authors do not actually know what is going on (a low probability event) in MST computation and certificates, please read Michel X. Goemans, David P. Williamson: A General Approximation Technique for Constrained Forest Problems. SIAM J. Comput. 24(2): 296-317 (1995) Incidentally, Definition 2 should not use the word "Steiner". Steiner trees are not defined in the manuscript, and nor are necessary.